# Performance Analysis of Localization Algorithms for Inspections in 2D and 3D Unstructured Environments Using 3D Laser Sensors and UAVs

**DOI:** 10.3390/s22145122

**Published:** 2022-07-07

**Authors:** Paul Espinosa Peralta, Marco Andrés Luna, Paloma de la Puente, Pascual Campoy, Hriday Bavle, Adrián Carrio, Christyan Cruz Ulloa

**Affiliations:** 1Centro de Automática y Robótica (CAR), Universidad Politécnica de Madrid (CSIC-UPM), 28006 Madrid, Spain; paul.espinosa.peralta@alumnos.upm.es (P.E.P.); marco.lunaa@alumnos.upm.es (M.A.L.); paloma.delapuente@upm.es (P.d.l.P.); adrian.carrio@dronomy.es (A.C.); christyan.cruz.ulloa@alumnos.upm.es (C.C.U.); 2Automation and Robotics Research Group, Interdisciplinary Centre for Security, Reliability and Trust, University of Luxembourg, 1855 Luxembourg, Luxembourg; hriday.bavle@uni.lu; 3Dronomy, Paseo de la Castellana 40, 28046 Madrid, Spain

**Keywords:** UAV, 3D sensor, particle filter, localization

## Abstract

One of the most relevant problems related to Unmanned Aerial Vehicle’s (UAV) autonomous navigation for industrial inspection is localization or pose estimation relative to significant elements of the environment. This paper analyzes two different approaches in this regard, focusing on its application to unstructured scenarios where objects of considerable size are present, such as a truck, a wind tower, an airplane, a building, etc. The presented methods require a previously developed Computer-Aided Design (CAD) model of the main object to be inspected. The first approach is based on an occupancy map built from a horizontal projection of this CAD model and the Adaptive Monte Carlo Localization (AMCL) algorithm to reach convergence by considering the likelihood field observation model between the 2D projection of 3D sensor data and the created map. The second approach uses a point cloud prior map of the 3D CAD model and a scan-matching algorithm based on the Iterative Closest Point Algorithm (ICP) and the Unscented Kalman Filter (UKF). The presented approaches have been extensively evaluated using simulated as well as previously recorded real flight data. We focus on aircraft inspection as a test example, but our results and conclusions can be directly extended to other applications. To support this assertion, a truck inspection has been performed. Our tests reflected that creating a 2D or 3D map from a standard CAD model and using a 3D laser scan on the created maps can optimize the processing time, resources and improve robustness. The techniques used to segment unexpected objects in 2D maps improved the performance of AMCL. In addition, we showed that moving around locations with relevant geometry after take-off when running AMCL enabled faster convergence and high accuracy. Hence, it could be used as an initial position estimation method for other localization algorithms. The ICP-NL method works well in environments with elements other than the object to inspect, but it can provide better results if some techniques to segment the new objects are applied. Furthermore, the proposed ICP-NL scan-matching method together with UKF performed faster, in a more robust manner, than NDT. Moreover, it is not affected by flight height. However, ICP-NL error may still be too high for applications requiring increased accuracy.

## 1. Introduction

Nowadays, industrial inspections using UAVs are common. In the last years, they have significantly improved in terms of flexibility, efficiency, less cost, etc., but more advances are necessary. Additionally, some of them are executed in dangerous and difficult environments for humans. Several tasks are executed in buildings [1], industries [2,3], power grids [4,5,6], agriculture areas [7,8,9,10], solar plants [11], and wind turbines [12,13]. Therefore, the development of autonomous robotic systems has become one of the most critical challenges in reducing human intervention in different types of missions.

Localization in these circumstances is often hard: the chosen method or algorithm must solve common problems such as initial positioning and kidnapping. During an autonomous inspection task, the dynamic nature of many unstructured environments is one of the major issues: objects can change their position, and new objects may appear in the scene. Updating the initial map for each new mission usually requires a lot of time and resources. In this context, the estimation of the robot pose considering the uncertainty in the sensors data and the vehicle dynamics within an unstructured environment is widely studied. A preliminary analysis of the best components must be performed to get good results and robust task execution.

This work’s main contribution is to adapt and analyze two different localization techniques for inspecting relevant objects in unstructured environments, which can help address most typical problems present in inspection scenarios. The first technique, based on AMCL, focuses on 2D localization, while the second technique, based on ICP and UKF, focuses on 3D localization. For each approach, a map is created in advance by different means. This article also studies different initialization options and convergence parameters. Conclusions from our experiences and results are drawn and highlighted in the article.

This paper is structured as follows: Section 2 collects and summarizes information from similar articles; in Section 3 procedures and algorithms are introduced. Then, the experimental setup, hardware, and results follow in Section 4, while the discussions are in Section 5. To conclude, Section 6 summarizes the main findings.

## 2. Related Work

Simultaneous Localization and Mapping (SLAM) [14] deals with the challenge of creating a map of an unknown environment and performing localization on that map at the same time. Robust SLAM methods are now available for mapping static, structured environments of limited size. However, performing SLAM in dynamic, large, and unstructured environments is still an open problem [15], so new methods are developed every year in order to gain more efficiency. Avola et al. [16] proposed a SLAM algorithm that provides the traveled route as well as the flight height by using both a calibration step and visual features extracted from the acquired images. The authors of [17], for instance, use two SLAM approaches, based on a 2D laser sensor and a monocular camera, to obtain logarithmic efficiency, compensating for the errors coming from them. Besides high computational cost, another important problem when applying online SLAM during task execution is robustness: it is less robust than localization on a previously validated map. Another approach [18] proposes a stereo SLAM method for UAVs to improve the robustness in rotating and motion scenes using two strategies, based on image quality and motion state.

Focusing on localization on a given map, several sensors or techniques may be combined with increased performance in some applications, but all of them have advantages and disadvantages. One of the most widely used pose estimation algorithms is the Extended Kalman Filter (EKF) [19]. It uses previous state information, control inputs, and sensor measurements. In [20], the authors present a work that uses a Global Position System (GPS) together with an Inertial Measurement Unit (IMU). Others, such as Mourikis et al. [21], combine visual characteristics from an omnidirectional camera and IMU data. The main limitation of this kind of method is the dependence on the initial position.

When the observation and state models are highly nonlinear, the UKF [22] is preferred since this filter calculates the pose using nonlinear models, resulting in a more accurate estimation than the EKF. In this regard, Ref. [23] made a comparison between UKF and EKF applied to low-cost sensors and very noisy measurements. Among the localization techniques that use nonlinear solutions for 3D environments, others are based on scan matching methods such as the Iterative Closest Point algorithm (ICP) and its variations [24,25,26], and others employ the Normal Transformation Distribution (NDT) [27,28,29], and some works, such as [30], use manifold Gaussian processes (mGPs) with geodesic kernel functions for mapping surface information to get potential spatial correlations and plan informative paths online in a receding horizon manner.

The Markov method [31] may be used when the initial pose is unknown. It works well with very noisy data, reducing ambiguities. This technique has some variants, including grid localization [32], where the map is divided into a grid, and each cell represents the occupancy probability. Another efficient and simple technique is Monte Carlo localization (MCL) [33]. This technique can solve global localization problems. Its principle is based on a particle filter distributed over the entire map with an estimated probability value. When the algorithm converges, the particle cloud’s size decreases. A variant called Adaptive Monte Carlo Localization (AMCL) [34] changes the number of samples over time instead of using a fixed-size particle cloud. This technique consumes less computational resources, is efficient, accurate, and able to solve the kidnapped robot problem. In this context, Xu et al. [35] proposed a hybrid localization method based on WiFi fingerprints and AMCL, Javierre et al. [36] combined a laser sensor and an omnidirectional camera, and [37] used a 2D/3D laser sensor with an ellipsoidal energy model.

For robot-assisted visual aircraft inspection, multiple approaches can be found in the literature, such as mobile platforms [38] or fixed cameras [39]. These methods simplify or avoid the localization problem, but their installation and operation could be more difficult than other solutions. Teleoperated climbing robots are able to perform autonomous inspections [40], but their navigation requires inputs from an operator. Other solutions present autonomous wheeled robots [41] and autonomous UAVs [42,43].

## 3. Methodology

In this work, we have chosen aircraft inspection as a test example because it takes place in an unstructured environment and aircraft positions may be different in each parking lot. Furthermore, the airplane geometry has several characteristics to be recognized, and the inspector UAV may take off from different positions. For this work, an open 3D CAD airplane model was used. We focus on aircraft inspection, but the selected methods and procedures would be similar in other inspection tasks, and other UAV models would also serve the same purpose. In addition, we have performed an inspection on a truck to test our methodology and compare the data acquired.

### 3.1. Maps

A map of the environment and an octomap (Figure 1a) [44] were built from the CAD model so as to test the performance of the algorithms. The octomap constitutes a probabilistic representation used to model unknown areas. It is efficient because it uses a tree structure called octree. Each node in an octree represents the space in a cubic volume, defined as a voxel. This volume is recursively subdivided into eight sub-volumes until a given minimum voxel size is reached.

An occupancy grid map [45] was built as a 2D map. It represents the environment as a block of cells, either occupied or unoccupied with some probability. To build this map, the octomap was projected into a horizontal plane (Figure 1b). The algorithm iterates over all octree leaves and marks the 2D map cells accordingly.

Another way to obtain a 3D map comes by applying SLAM. We selected a graph SLAM [46] algorithm, which uses a graph whose nodes correspond to the robot’s poses at different time points and whose edges represent constraints between the poses (Figure 1c). Once such a graph is constructed, the map can be computed by finding the nodes’ spatial configuration that is most consistent with the measurements modeled by the edges.

### 3.2. 2D Localization

Localization in 2D is performed by means of the AMCL [47] algorithm. A particle filter integrates a robot motion model, an observation model, and an adaptive algorithm that adds random particles based on its convergence; Figure 2 and Appendix A show an example of the algorithm performance. The following relation can summarize it:(1)$$X=f(X_{t-1},U_{t},Z_{t},m)$$
where:*X*, current set of particles.$X_{t-1}$, the previous set of particles.$U_{t}$, last motion and odometry measurements.$Z_{t}$, last laser rangefinder measurements.*m*, max number of particles.

In addition, there are several aspects to consider, such as:Motion kinematics: e.g., differential or omnidirectional.Uncertainty of the robot odometry, which determines the error in translation $\alpha_{1},\alpha_{2}$ or rotation $\alpha_{3},\alpha_{4},\alpha_{5}$.Observation model: e.g., beam model or probability field model.Measurements errors: e.g., measurement noise $z_{hit}$, unexpected objects $z_{short}$, object detection failures $z_{max}$, unexplained random noise $z_{rand}$.Number of random particles, defined by the probabilities of long-term $\alpha_{slow}$ and short-term $\alpha_{fast}$ measurements.

Nowadays, there is no general method other than calibration or trial and error to set the parameters of motion model noise, measurement model noise, and particle decay rate. Assigning values without any previous experience is very complicated.

The motion model selected in this work corresponds to Omnidirectional dynamics, typical of UAV’s motion (DJI Matrice 100 and hummingbird UAV). The DJI Matrice 100 has very accurate odometry samples, and the error is minimal, so the values related to the noise of the motion model are small. The error values in simulated odometry sensors mounted on the Hummingbird UAV were based on the default values.

The observation model is the Probability field measurement model because of its lower computational cost and its smooth error distributions. It is also known that the Velodyne laser sensor does not present many errors in terms of reflection, maximum distance, and loss of object detection. The main source of error is the measurement noise itself, so $z_{hit}$ should have a large value, $z_{rand}$ a small value, and $z_{short}$; $z_{max}$ much smaller. The AMCL simulation test parameters are given in Appendix D, and those of AMCL real flights are presented in Appendix E.

AMCL estimates only the position of x and y, and the z-coordinate is always set to zero because the algorithm assumes that all objects are on the horizontal surface. Therefore, flights must be performed at some horizontal and vertical distance from the aircraft to guarantee that the 3D laser detects complete point clouds which present a correct projection. The horizontal distance between the UAV and the aircraft is between 2 and 3 m, depending on the inspection trajectory (Figure 3). Moreover, some flights at various altitudes were performed to find the best altitude.

Several flights were executed. The take-off position and the initial uncertainty were random because we wanted to test how the system would work with arbitrary large errors in the initial pose estimation. Trajectories with the smallest error were selected, and covariance values given by AMCL cov_x, cov_xy, cov_y were recorded, as well as the number of occupied cells observed by the laser. These values give an indication of convergence and error. These results were used in real flights to determine a convergence position in the future; it could be used as an initial position to execute another inspection task.

### 3.3. 3D Localization

Three-dimensional localization is performed by adapting the algorithm presented in [29] for NDT and IMU data with UKF. The algorithm used in this work replaces the proposed scan-matching techniques: NDT, ICP, and Generalized ICP by nonlinear ICP [48]. Figure 4 and Appendix A show an example of the algorithm’s performance.

The algorithm is based on three stages:Subsampling: Reduction of the number of points in the sample using Voxel grid filter and Pass-through Filter techniques [49] to optimize processing time.Pose estimation with nonlinear ICP: Based on the ICP technique presented in [50]. It takes as inputs a source and a target point cloud matched under the nearest-neighbor criterion. Singular Value Decomposition (SVD) is applied to obtain an estimate of the transformation matrix that aligns them. This process is repeated until a termination criterion is met, removing outliers and redefining the correspondences.Unscented Kalman Filter-based localization: This is an improvement of EKF for application to highly nonlinear systems. This approach uses the unscented transform to take a set of samples called sigma points, which are propagated by nonlinear functions and used to calculate the mean and variance. Unlike EKF, UKF eliminates the need for a Jacobian, facilitating calculations on complex functions.

To obtain the 3D pose, the algorithm takes as inputs: the 3D point cloud map of the environment, the previous state including position and orientation, LIDAR sensor data, and IMU data together with linear and angular velocities. The result is the 3D pose. First, the map and LIDAR sensor data are subsampled. Next, pose estimation is performed based on the IMU linear and angular accelerations within a defined time frame. These data are used in the filter prediction stage; then, the scan matching algorithm performs the pose estimation by aligning the map point cloud and the sensor point cloud. This estimation will be used for the filter correction stage.

The pseudocode for this solution is presented in Algorithm 1:
**Algorithm 1:** Localization 3D ($X_{t-1}$, $V_{t}$, $\Delta_{qt}$, 3D_sensor_points, 3D_map)
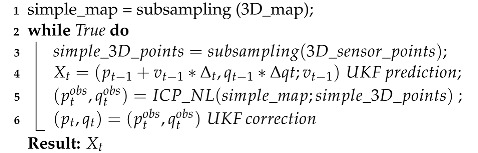


To test the performance of the ICP-NL and NDT algorithms in simulation, the trajectory of Figure 3 was executed at different heights, and the ATE and convergence time was calculated. In addition, the coincidence between the 3D laser sensor point cloud and the generated 3D map is qualitatively verified in simulation and real data, and the deviations between the estimated 3D position and the real position are also observed.

## 4. Results

Initially, we specify the software and hardware used, as well as the metric to determine the trajectory estimation errors. Then the 2D localization results in simulation and real flights, and finally, the 3D localization results.

### 4.1. Software

This work was developed by using the popular Robot Operating System (ROS) [51] and Gazebo [52] simulation tools, which help create simulations of robotic applications in three-dimensional indoor and outdoor environments. The Aerostack framework [53], which allows developers to design and test control architectures for aerial robotic systems, was also extensively used. Rotors [54] provided the simulation of UAVs and localization sensors. Figure 5 shows the complete software system used to perform the localization task in simulations and in real flights.

### 4.2. Hardware

About the actual UAV assembly, the hardware components used in simulation flights are as follows:Personal Computer (PC) Intel core i7 2.70 GHz, 4 Gb RAM.Hummingbird UAV provided by Rotors (Figure 6a).Odometry measurements from a configurable odometry sensor (see Appendix B for configuration details).Velodyne VLP16 3D laser sensor [55] provides a point cloud with 300,000 points per second and ±3 cm accuracy, 100 m range, 360° horizontal and 30° vertical field of view (see Appendix C for setting other parameters). Due to computational limitations in the simulation, we worked with 5120 points per second.

In real flights, we used:Manifold Intel i7 1.8 Ghz, 2 GB RAM.DJI Matrice 100 UAV (Figure 6c).Odometry measurements were provided by a sensor fusion algorithm [56] that merges the onboard DJI sensors: Altimeter, Velocity and IMU, plus the 3D Light Detection and Ranging (LIDAR) IMU.3D LIDAR Ouster OS1-64, providing a point cloud with 327,680 points per second and ±1 cm accuracy 100 m range, 0.3 cm resolution, 360° horizontal and 45° vertical field of view.

In both setups, we used the Airbus A330 3D CAD model [57] to build the prior maps. The teleoperation flights and trajectory tracking were controlled by Aerostack.

### 4.3. Error Metric

The Absolute Trajectory Error (ATE) is a measure to check that the solution to a benchmark problem is valid. ATE is applicable in algorithms for robot trajectory estimation. It measures the difference between the ground-truth position and the robot’s estimated position at the same instant in time. Further, it should be noted that the robot pose’s orientation component is not considered [59,60]. In this work, ATE is only computed in simulation environments due to a lack of ground-truth data in real environments.
(2)$$ATE=\sqrt{\frac{1}{n}\sum_{i=1}^{n}T_{i}^{2}}$$
where:$T_{i}$, is the magnitude of the Euclidean distance along the horizontal plane between the estimated and ground-truth poses at frame *i*.*n*, number of frames.

### 4.4. 2D Localization

To build the 2D occupancy grid map, a 3D CAD model in Digital Asset Exchange (DAE) format for Gazebo was imported (Figure 7a). An octomap of the environment was created with a 0.1 m voxel resolution (Figure 7b), and then it was projected on a horizontal flat plane to make a 2D occupancy map with a 0.1 m cell resolution (Figure 7c).

Subsequently, this map was modified by an external script to obtain the image’s edges. This could improve the algorithm’s performance because the black pixels represent an occupied area on the map. When comparing the map with the laser measurements, the algorithm could incorrectly interpret the internal occupied cells and give a wrong estimation of the position. The new 2D map is shown in Figure 7d.

#### 4.4.1. Simulation Tests

Regarding localization performance, the AMCL parameters were calibrated in the simulation to verify the result’s quality. When the UAV takes off, the particle distribution is very sparse, as shown in Figure 8a. As the UAV advances, the particle distribution gets more compact, as shown in Figure 8b. When the UAV keeps flying longer, the algorithm converges, and the 2D laser scan matches the map, as shown in Figure 8c. For a complete visualization, please refer to Appendix A.

To find the best flight altitude in combination with the inspection trajectory (Figure 3) and 3D laser scan configuration (range [0.1, 100] meters; horizontal and vertical field of view respectively, [−360, 360] degrees, [−15, 15] degrees) 20 flights were executed in simulation at each altitude. Table 1 shows the result, where 5 m corresponds to the altitude with the lowest ATE. Therefore, 5 m is the altitude executed in all simulated flights to evaluate 2D performance.

To evaluate the performance, 81 different flights were tested. The proposed initial error was three meters maximum for each axis reference (x,y). Each flight starts at a random initial position taken from the inspection trajectory (Figure 3) and presents an initial error not always smaller than three meters, so in some cases, the algorithm may not converge. Table 2 presents the average, minimum, and maximum ATE errors. Figure 9 presents the trajectory with the lowest error, and Figure 10 the trajectory with the highest error; in this case, the algorithm does not converge because the initial covariance is smaller than the error in the initial position.

Table 3 shows the initial position and the ground truth (GT) position, with the maximum and minimum ATE. When the difference between the GT and AMCL pose is large, the algorithm starts from erroneous estimates. If the initial covariance is large enough regarding the error in the initial position, the algorithm should converge.

The execution of a short flight before starting the inspection can help improve localization in these kindd of tasks, providing the resulting pose as the new initial position. It reduces the uncertainty in the initial position and allows for correct position estimation and a safe flight. Two types of trajectories have been defined in this work, a circular trajectory (Figure 11b) and a squared trajectory (Figure 11a). A total of 91 flights were performed with each trajectory and different initial positions (red point in Figure 11) along the inspection trajectory (Figure 3). The analysis of the results indicates that the squared trajectory worked best, with a mean ATE of 2.429 (m).

To further analyze the results, the particle covariance matrix in the x and y directions was obtained. We also logged the number of matches between occupied cells and 2D laser scan points for those tests with an initial error lower than 0.3 m. Table 4 shows these values.

We realized other simulations to compare the results. We created a 2D occupancy grid map from a 3D CAD truck model using the same procedure explained in Section 4.4. Figure 12 shows the different steps of the map-building process.

Thirty-six different flights were performed using the same aircraft simulation parameters, each flight starting at a different point around the trajectory proposed in Figure 13. Figure 14 shows the flight with the lowest error, and the first row in Table 5 shows the initial parameters. In this case, there is a considerable AMCL pose estimation error at the beginning, but when the UAV continues flying, the algorithm converges until reaching some complicated truck place. Appendix A shows the full flight. Figure 15 shows the flight with the highest ATE, and the second row of Table 5 shows the initial parameters. In this flight, the initial error is the lowest. Even though AMCL does not converge along some parts of the trajectory, most of the time, the algorithm estimates the correct pose.

The tests performed allowed data indicating when the algorithm converges to be obtained. We take all the points of the flights with an error of less than 30 cm and calculate the mean of the particle covariance matrix in x and y, the number of matches between occupied cells, and the 2D laser scan point. Data can be seen in Table 6.

#### 4.4.2. Real Tests

To collect data in a real environment, the UAV pilot performed several flights where the object to inspect had relevant characteristics; in each flight, we recorded information from the UAV such as altitude, velocity, IMU, and the point cloud provided by a 3D laser in rosbag files. Then we played rosbag files with the same timestamp in all topics to get the saved information.

Three real, short flights were chosen before starting the inspection to check the algorithm’s convergence. The proposed trajectories are shown in Figure 16. In real tests, stairs are present in 3D laser scan data Figure 17a. Therefore, stairs were added to the initial 2D occupancy grid map to better represent the environment in Figure 17b.

In the real flights, walls were not considered, and this had a negative effect on the algorithm performance. To solve this problem, the first option was the method proposed in [61], based on plane segmentation by means of RANSAC [62] and the implementation available in the Point Cloud Library (PCL) [63]. Figure 18a shows the 3D Point Cloud and the original conversion to a 2D laser scan; Figure 18b shows the wall segmentation and the projection to a 2D laser scan. The second option is to limit the 2D laser scan data in range and angle of view, as shown in Figure 18c.

The first proposed trajectory was tested with a laser range limitation of 15 m and an angular field of view restricted to [−135, 135] degrees. Figure 19 shows that the particles are very spread at the beginning of the trajectory. When the UAV moves, the scattering decreases, the laser matches the map, and the covariance in x and y decreases below the desired values. In this case, 37 iterations were required to converge. Figure 20 shows the results using wall segmentation for RANSAC PCL, which required 250 iterations to converge, and Appendix A shows the complete flight.

Wall segmentation with RANSAC PCL tests has been performed following Path 2 (Figure 16b) and Path 3 (Figure 16c) because the trajectory is close to the nose of the aircraft, and the laser does not capture enough information of the whole environment. Figure 21 and Appendix A show the Path 2 result, which converged in 71 iterations. As for Path 3, the results are shown in Figure 22. In this case, the algorithm was not able to converge, so the trajectory is not good enough to achieve an acceptable initial position.

### 4.5. 3D Localization

#### 4.5.1. Simulation Tests

The 3D map was built based on the graph SLAM algorithm presented in [27,64]. This algorithm uses odometry obtained from NDT-scan matching and loop closure (detection of an area that has already been mapped) with corrections based on GPS data. In this work, this method was simplified by replacing the odometry based on scan-matching with the robot’s real position obtained from the simulation environment, obtaining higher accuracy in the map. Figure 23 below shows the 3D map obtained.

The NDT and ICP-NL algorithms have been compared to observe their performance and differences. For this purpose, the ATE has been calculated for the flights performed. The results are presented in Table 7. Figure 24 shows the main results, three stages of operation: starting position, take-off, and navigation, were performed. The trajectories painted in white represent the algorithm estimation, and the trajectories painted in green represent the real movement.

The estimated initial position for the two cases differs by 5 m in x and y from the real one. When using NDT, the algorithm tries to estimate a trajectory, but it fails to converge to the actual pose (Figure 24b). In ICP-NL, the algorithm converges during take-off because the sensor observes more points, and during navigation, the error remains low (Figure 24a). For a complete visualization, please refer to Appendix A.

Additionally, several flights were performed by modifying the UAV flying height between 3.5 and 8.5 m in order to study the influence on the results. We observed that ICP-NL maintains its performance regardless of the altitude. On the other hand, NDT works well in the last test altitudes because the higher the altitude, the better the aircraft is observed, and hence the scan-matching algorithm performs better. Considering the initial error established, the established convergence time is the time it takes for each method to minimize the error in Euclidean distance to less than 1m. It was observed that the proposed ICP-NL method was much faster at converging. The height errors and convergence times are presented in Table 8.

The simulated flight with the lowest ATE and the shortest convergence time, corresponding to a height of 7.5 m, was taken as a reference. Figure 25a shows the trajectories generated by the ICP-NL algorithm and the ground truth. The two trajectories are practically overlapped along the whole path, and the ATE and ABS errors in Figure 25b are within the margins allowed in the application. This method provides very good results in the simulation. Figure 26 shows the flight with the highest error.

Like the 2D case, we evaluate the performance of the ICP-NL using the 3D CAD truck model. The result of the 3D map using the graph SLAM approach is shown in Figure 27, and Appendix A shows how a 3D map is built when the UAV is moving.

Based on previous results, only the ICP-NL approach was evaluated. Table 9 presents the results at different heights.

Different flights were performed using the same aircraft 3D simulation parameters; each flight started at a different initial position with an initial error of less than one meter. Figure 28 and  Figure 29 show the flight with the lowest ATE, and Appendix A shows the pose estimation by ICP-NL during the flight. Further, Figure 30 shows the flight with the highest ATE.

#### 4.5.2. Real Tests

To check the method’s effectiveness, tests were performed with real data as well, using the same files (ROS rosbags) as in the AMCL algorithm tests. Figure 31a and Appendix A correspond to the trajectory of flight Figure 16a and Figure 31b and Appendix A correspond to flight Figure 16b. In Figure 31, the odometry trajectories have been represented in white. The algorithm’s correct operation can be checked visually if the data captured by the sensor overlaps with the reference map’s data and if the trajectory generated by the odometry is close to the trajectory provided by ICP-NL. However, in this case, the trajectory estimated by ICP-NL has many deviations, and the 3D map data does not match the 3D laser data. The main cause of these errors when evaluating the real environment methods is assumed to be the noise generated by the walls and other unexpected objects adjacent to the aircraft.

## 5. Discussions

### 5.1. 2D Localization

Using a static 2D map built from a standard CAD model has significant advantages over developing SLAM algorithms to map the environment, as it can save time and resources and provide enhanced robustness. Furthermore, using a 3D laser sensor for 2D environments is very practical, can be cost-effective, and can produce really nice results, as shown in this work.

Unexpected objects not present on the map can be segmented in a simple manner by modifying the laser data range. These parameters must be optimized according to the application and environment to obtain the desired results. Object segmentation can also be performed by applying other typical point cloud processing algorithms, if necessary.

If the map contains objects with particular geometries or noticeable corners, the detection of relevant features is evident. As more data are acquired, the filter will converge easily. In this work, the wings’ intersection with the aircraft’s body and the corners of the truck provides a unique feature. This area can be used to detect the relevant features. It provides unique features and, therefore, the area around can be used as a take-off point for the algorithm to converge faster.

When using AMCL, a considerable degree of uncertainty in the initial position is admissible if safe navigation is guaranteed, as the algorithm can estimate the location accurately after some time. For a flight to be robust and accurate, AMCL can be used in two stages. The first one aims to determine its initial position with minimal error by performing a specific trajectory and waiting for the algorithm to converge. The second stage starts the desired flight considering the initial estimated position. As expected, the lower error in AMCL initialization, the better the results in estimating the whole trajectory. Further, AMCL can be used to determine a reliable initial position with little error for other algorithms in which this information is critical, such as the EKF.

The results obtained from the truck inspection show that the covariance XY, laser data, and occupied cells are lower than the values from the aircraft inspection. Because the 2D map created and the truck is smaller, the data obtained by each component has less information.

### 5.2. 3D Localization

Regarding mapping, the graph SLAM changes applied for this work were adequate. The obtained map had little noise, which helped rule out sources of error in the localization algorithms execution. Additionally, the point cloud map could be sub-sampled to optimize processing time. In the simulations, we observed some false positives in pose estimation. Therefore, it is recommended to build the 3D map without using the floor in the world.

Despite the noise generated by elements other than the aircraft, we showed that a map generated in simulation could be used for localization in a real environment with nice results. However, it would be an exciting contribution to test the algorithms in an open field or far from noisy elements to corroborate their better operation.

Overall, the proposed ICP-NL scan-matching method and UKF performed better than NDT in the performed tests. Its behavior was robust, and its results were not affected by parameters such as flight height. Moreover, its convergence time was relatively short despite estimation errors at the initial position. However, the error may still be too high for applications requiring increased accuracy.

NDT achieves high accuracy in the estimation of the x and y coordinates in some of the flights. In the case of NDT, it works very well as long as the initial uncertainty does not grow too much. Otherwise, it has significant costs to recover.

According to the performed tests, nonlinear ICP supports initial positions with a more extensive range of uncertainty than similar algorithms such as NDT, ICP, and Generalized ICP.

## 6. Conclusions

### 6.1. 2D Localization

Performing 2D localization on a pre-built map using a CAD model and a 3D laser sensor is very practical. It saves resources and time, and the results are good.

Unexpected objects inside the environment may be deleted in a simple and fast manner by imposing 2D laser range limitations. However, the results can be improved by using PCL segmentation methods.

The initial position used in AMCL could have a considerable degree of uncertainty. Places with relevant features on the map should be used for the initial position to reduce convergence time and improve the quality of localization.

The position estimated by AMCL can be used as an initial position for other algorithms.

### 6.2. 3D Localization

Applying sub-sampling methods to the point cloud can reduce the time to build a map. To obtain higher accuracy in the 3D map, the robot’s real position may be fed to the algorithm proposed by [27,64].

In our tests, the best results were obtained by the ICP-NL scan-matching method together with UKF. It is a robust algorithm; flight height does not affect its performance, it works well with an extensive range of uncertainty, and its convergence time is relatively short. NDT had a good performance in pose estimation in x and y with a long-range of uncertainty. However, in some cases, it presented failures, and the robot got lost.

## Figures and Tables

**Figure 1 sensors-22-05122-f001:**
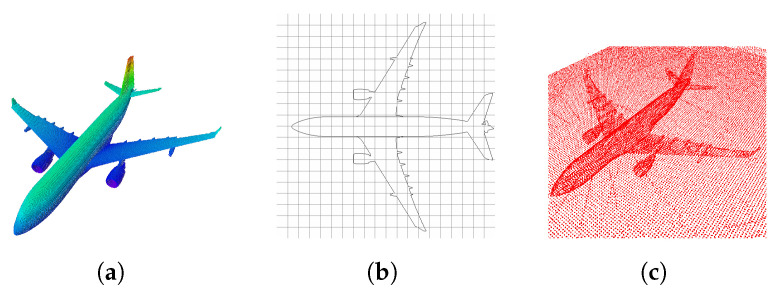
Kind of maps to be used. (**a**) Three-dimensional Occupancy grid Map (Octomap). (**b**) Two-dimensional Occupancy grid Map. (**c**) Graph SLAM map.

**Figure 2 sensors-22-05122-f002:**
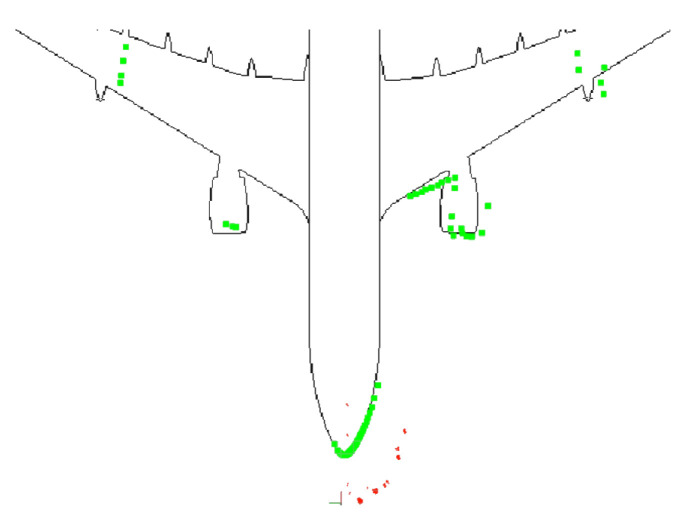
Example of 2D localization using AMCL algorithm (red), 2D laser Scan (green) and an Occupancy grid map (black).

**Figure 3 sensors-22-05122-f003:**
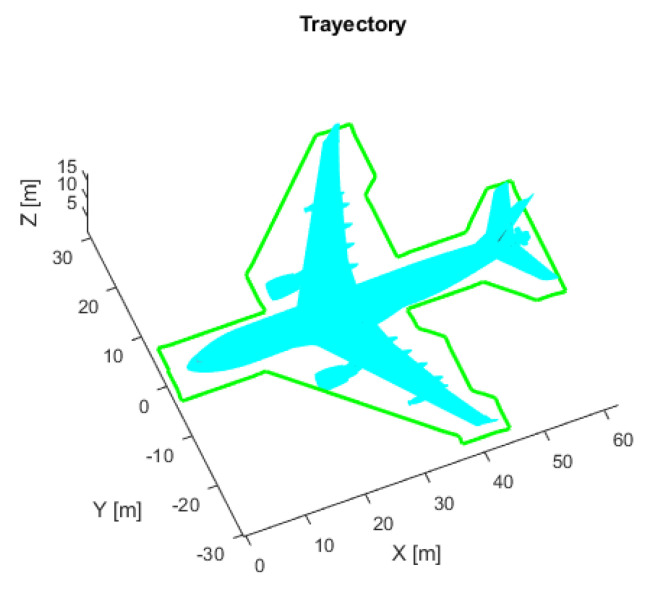
Proposed inspection trajectory (green). Three-dimensional CAD model (cyan).

**Figure 4 sensors-22-05122-f004:**
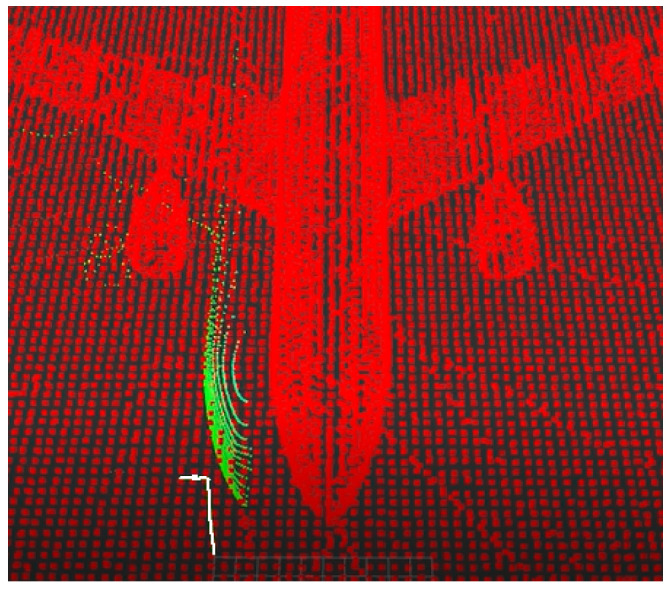
Example of 3D localization using nonlinear ICP algorithm (white), 3D laser Scan (green) and a 3D Occupancy grid map (red).

**Figure 5 sensors-22-05122-f005:**
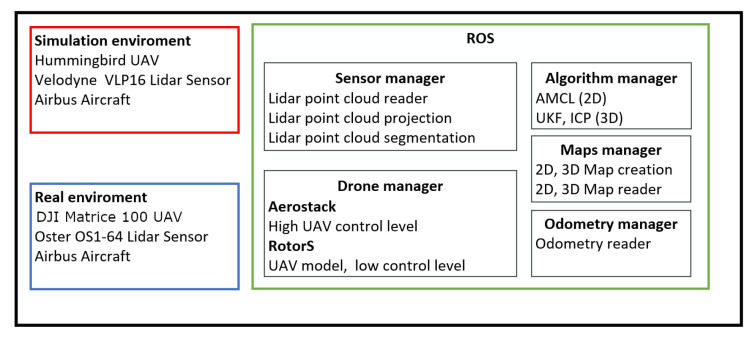
Proposed system architecture for simulations and real flights.

**Figure 6 sensors-22-05122-f006:**
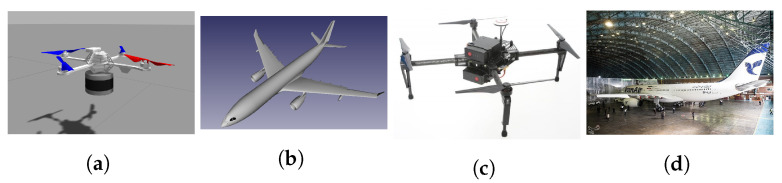
Components used in the simulation (**a**,**b**), and real flights (**c**,**d**). (**a**) Velodyne sensor and Hummingbird UAV. (**b**) Aircraft Airbus A330 model. (**c**) Three-dimensional LIDAR Ouster and DJI Matrice 100 UAV. (**d**) Airbus A330 [58].

**Figure 7 sensors-22-05122-f007:**
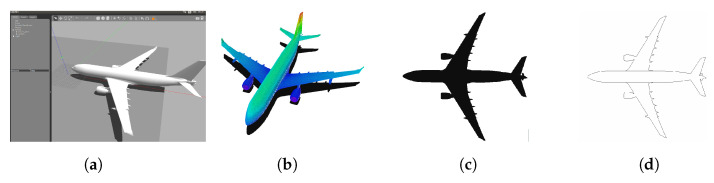
2D occupancy grip map creation process. (**a**) Airbus A330 Gazebo Model. (**b**) Octomap generated with 0.1 cm voxel resolution. (**c**) Octomap horizontal projection. (**d**) Modified 2D occupancy grip map.

**Figure 8 sensors-22-05122-f008:**
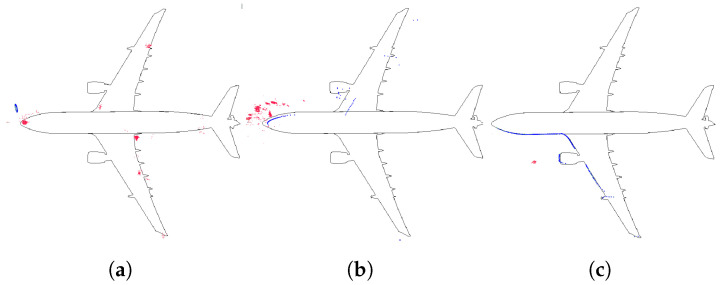
AMCL convergence process. Two-dimensional laser scan (blue), AMCL particle cloud (red). (**a**) UAV takes off, particles are distributed all over the map. (**b**) UAV moves forward, particle scattering begins to decrease The 2D laser scan does not match the map. (**c**) AMCL converges, particle scattering is small, and the laser data matches the map.

**Figure 9 sensors-22-05122-f009:**
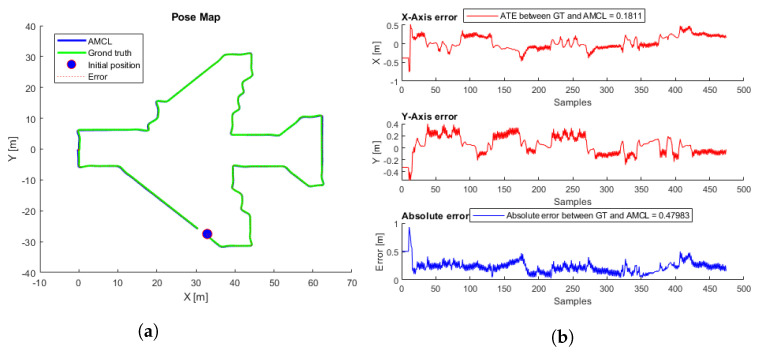
(**a**) AMCL simulation test with the lowest ATE. AMCL (blue), ground-truth (green), take-off position (blue circle), error (red). (**b**) AMCL errors.

**Figure 10 sensors-22-05122-f010:**
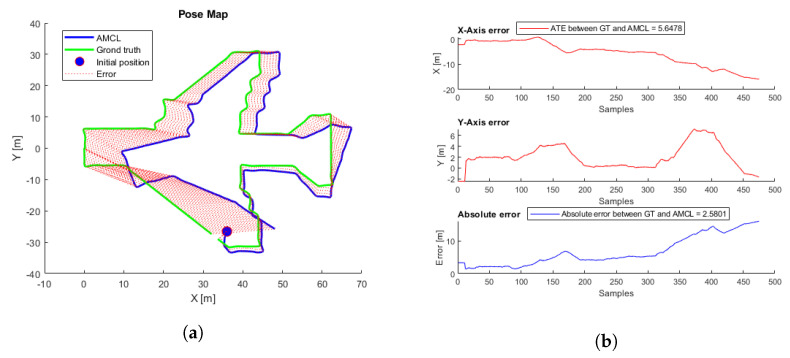
(**a**) AMCL simulation test with the highest ATE. AMCL (blue), ground-truth (green), take-off position (blue circle), error (red). (**b**) AMCL errors.

**Figure 11 sensors-22-05122-f011:**
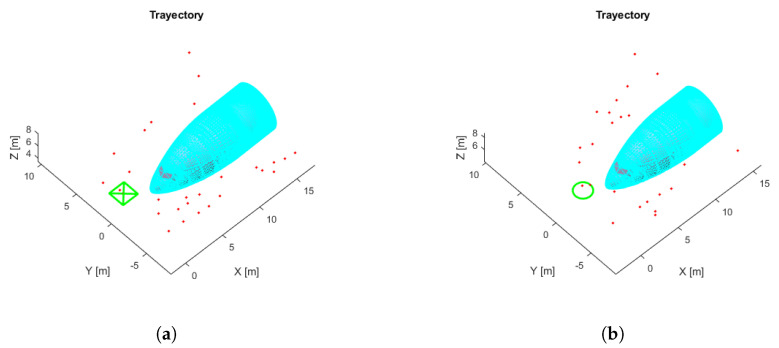
Proposed initial flight (green), Aircraft (cyan), take-off point (red). (**a**) Squared path. (**b**) Circular path.

**Figure 12 sensors-22-05122-f012:**
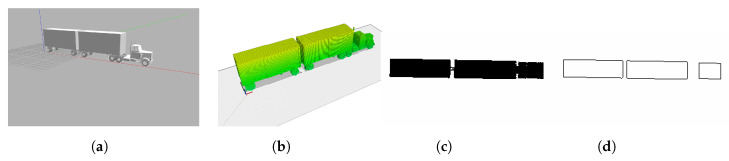
Two-dimensional occupancy grip map creation process. (**a**) Truck Gazebo Model. (**b**) Octomap generated with 0.1 cm voxel resolution. (**c**) Octomap horizontal projection. (**d**) Modified 2D occupancy grip map.

**Figure 13 sensors-22-05122-f013:**
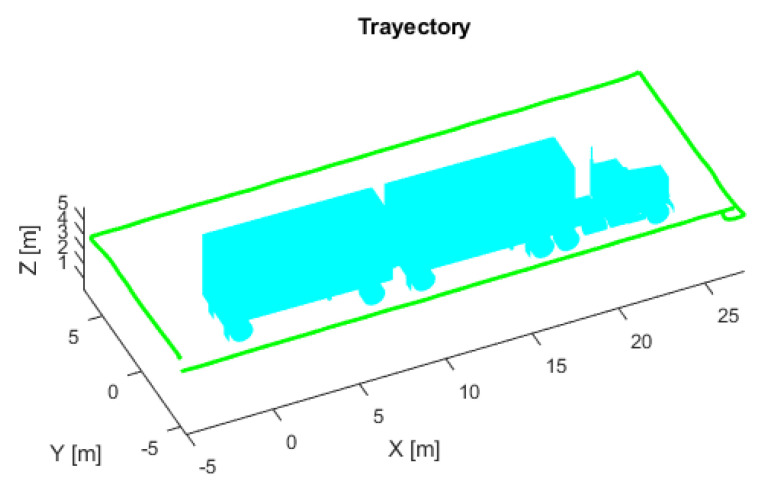
Proposed inspection trajectory (green). Truck 3D CAD model (cyan).

**Figure 14 sensors-22-05122-f014:**
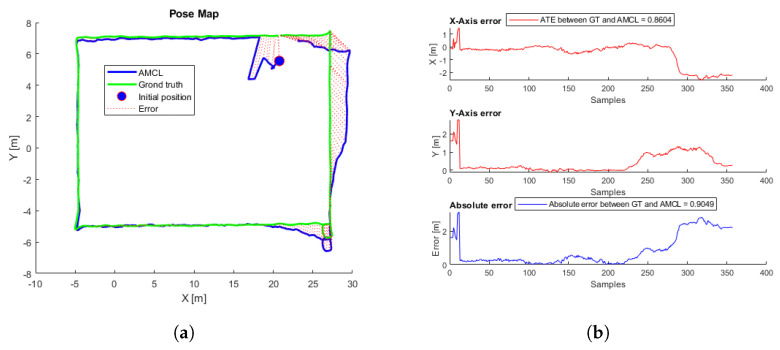
(**a**) AMCL simulation test with the lowest ATE. AMCL (blue), ground truth (green), take-off position (blue circle), error (red). (**b**) AMCL errors.

**Figure 15 sensors-22-05122-f015:**
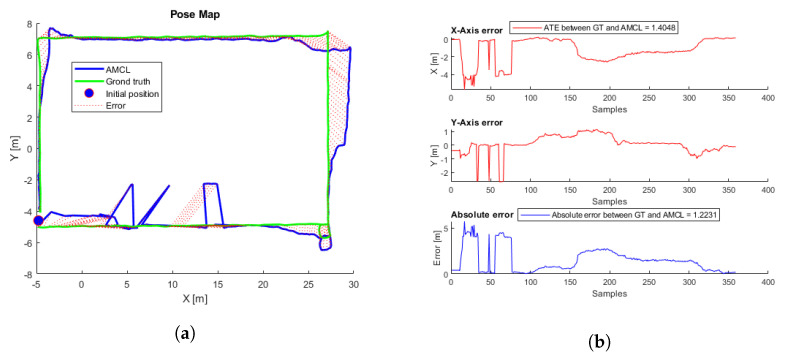
(**a**) AMCL simulation test with the highest ATE. AMCL (blue), ground truth (green), take-off position (blue circle), error (red). (**b**) AMCL errors.

**Figure 16 sensors-22-05122-f016:**
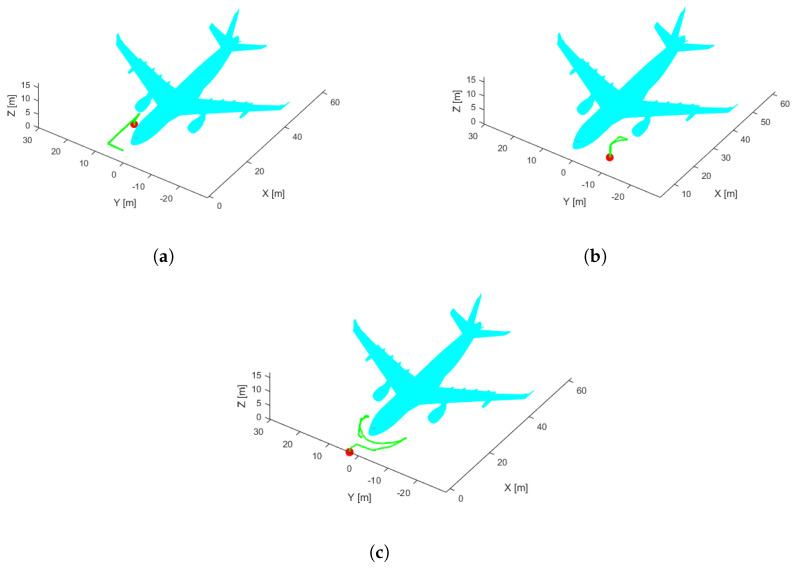
Proposed trajectories for real flights. Take-off and landing point (red circle). (**a**) Path 1. (**b**) Path 2. (**c**) Path 3.

**Figure 17 sensors-22-05122-f017:**
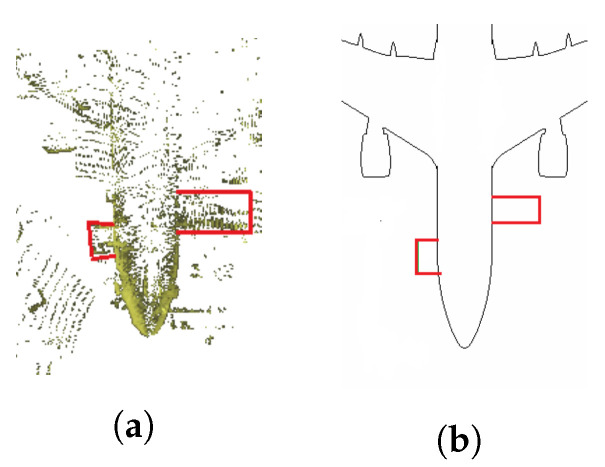
Map update, stairs (red). (**a**) 3D Laser scan real data. (**b**) New occupancy grip map.

**Figure 18 sensors-22-05122-f018:**
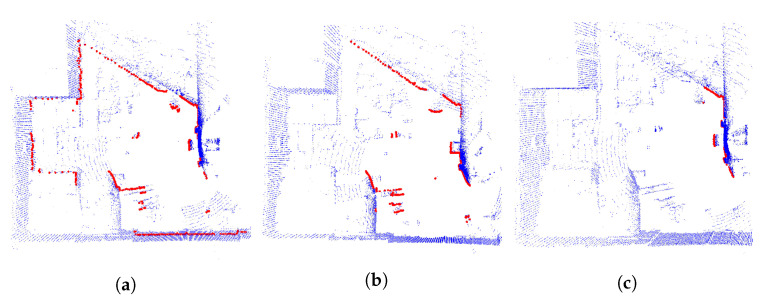
Wall removal. Three-dimensional LIDAR point Cloud (blue), 2D laser scan (red). (**a**) Original 2D laser scan data and 3D LIDAR point cloud. (**b**) Wall segmentation by RANSAC PCL. (**c**) Wall segmentation by limiting 2D laser scan data. Range [0.1, 15] meters, angular field of view [−135, 135] degrees.

**Figure 19 sensors-22-05122-f019:**
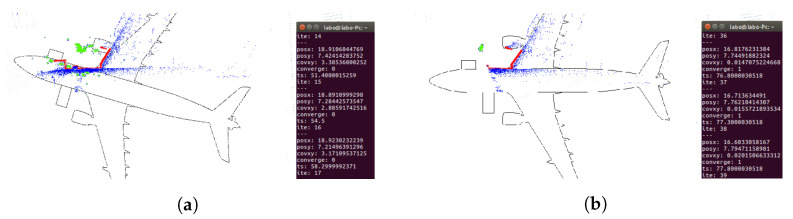
Convergence tests on Path 1 (Figure 16a) real flight with limited laser ranges. Three-dimensional LIDAR point cloud (blue), 2D laser scans (red), AMCL particles (green). (**a**) UAV position before algorithm convergence. (**b**) UAV position when the algorithm converged at 37 iterations.

**Figure 20 sensors-22-05122-f020:**
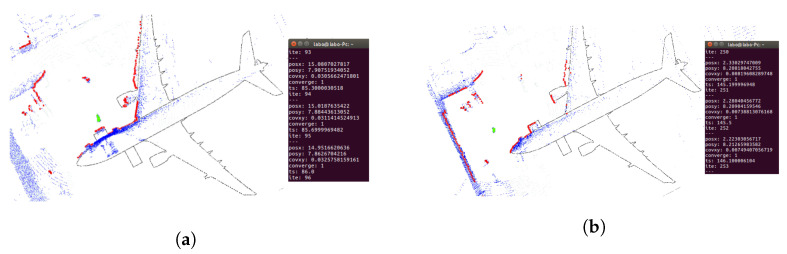
Convergence tests on Path 1 (Figure 16a) real flight with RANSAC PCL segmentation. Three-dimensional LIDAR point cloud (blue), 2D laser scans (red), AMCL particles (green). (**a**) UAV position before algorithm convergence. (**b**) UAV position when the algorithm converges after 250 iterations.

**Figure 21 sensors-22-05122-f021:**
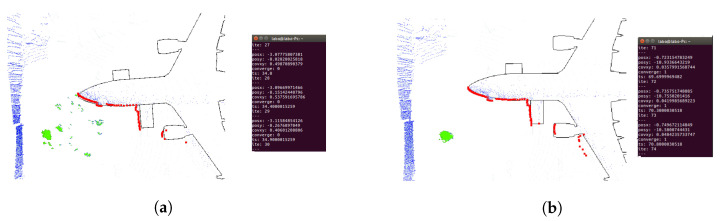
Convergence tests on Path 2 (Figure 16b) real flight with RANSAC PCL segmentation. Three-dimensional LIDAR point cloud (blue), 2D laser scans (red), AMCL particles (green). (**a**) UAV position before algorithm convergence. (**b**) UAV position when the algorithm converges after 71 iterations.

**Figure 22 sensors-22-05122-f022:**
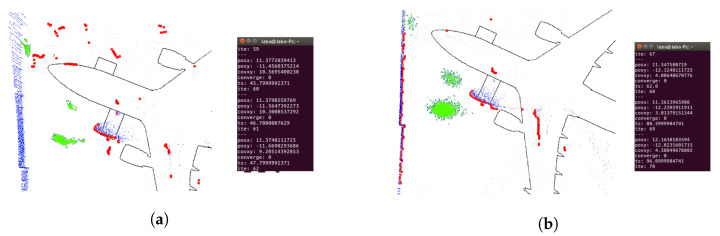
Convergence tests on Path 3 (Figure 16c) real flight with RANSAC PCL segmentation. Three-dimensional LIDAR point cloud (blue), 2D laser scans (red), AMCL particles (green). (**a**) UAV position before algorithm convergence. (**b**) The algorithm does not converge during the flight.

**Figure 23 sensors-22-05122-f023:**
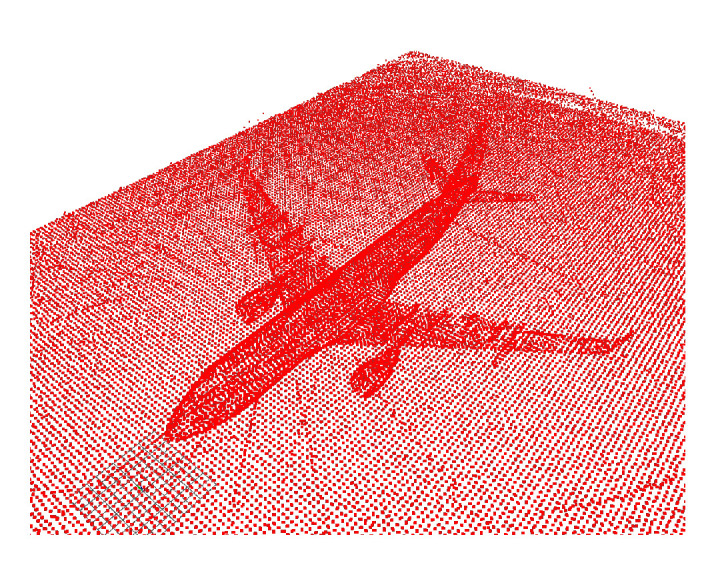
Aircraft 3D map generated by graph SLAM.

**Figure 24 sensors-22-05122-f024:**
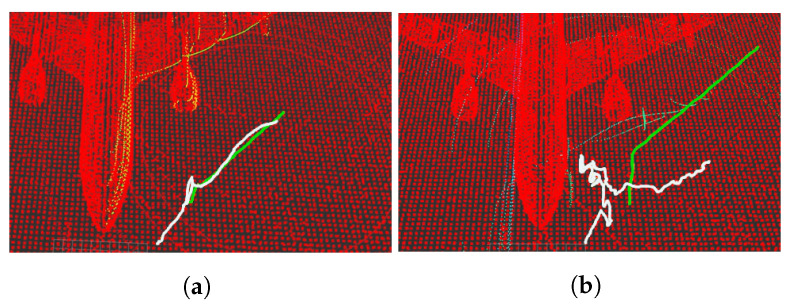
Position estimate by the algorithms Non-linear ICP (**a**) and NDT (**b**). Three-dimensional graph SLAM map (red), position estimate by the algorithm (white), ground truth (green).

**Figure 25 sensors-22-05122-f025:**
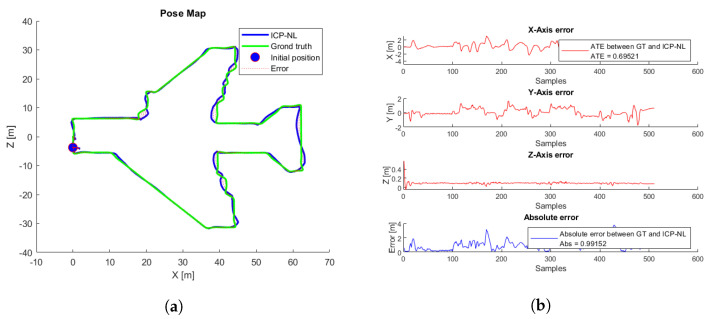
ICP-NL simulation test with the lowest ATE. (**a**) Trajectories. ICP-N (blue), ground truth (green), take-off position (blue circle), error (red). (**b**) ICP-N errors.

**Figure 26 sensors-22-05122-f026:**
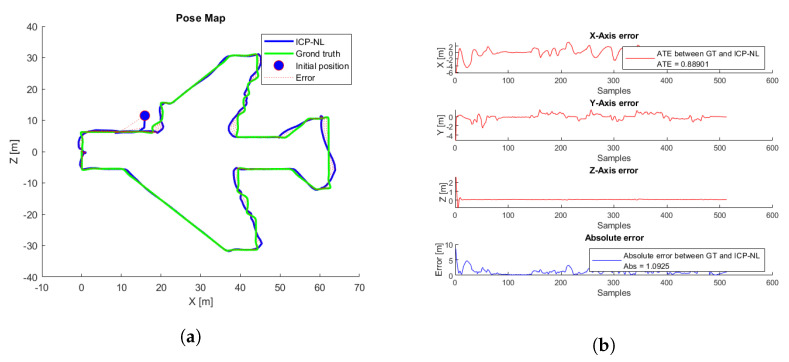
ICP-NL simulation test with the highest ATE. (**a**) Trajectories. ICP-NL (blue), ground truth (green), take-off position (blue circle), error (red). (**b**) ICP-N errors.

**Figure 27 sensors-22-05122-f027:**
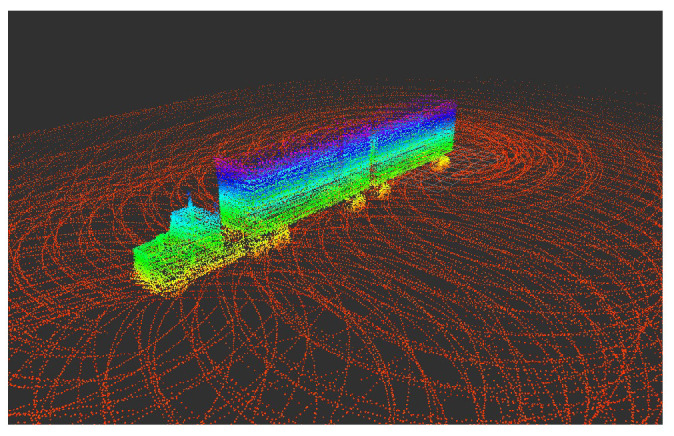
Truck 3D map generated by graph SLAM.

**Figure 28 sensors-22-05122-f028:**
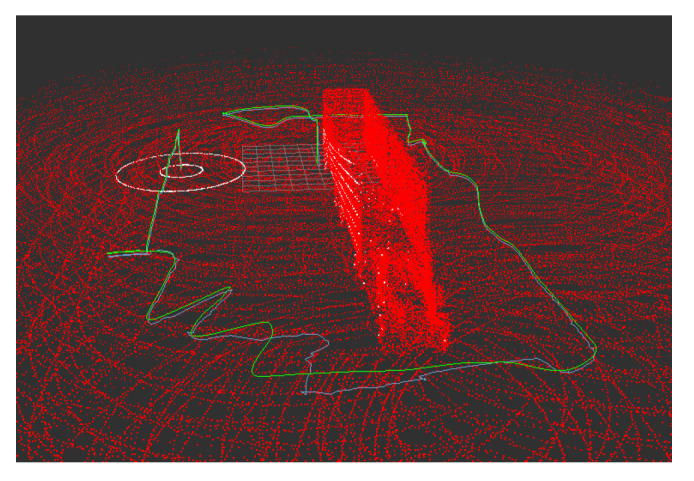
Pose estimation by ICP-NL. Three-dimensional map (red), ground truth path (green), ICP-NL path (blue).

**Figure 29 sensors-22-05122-f029:**
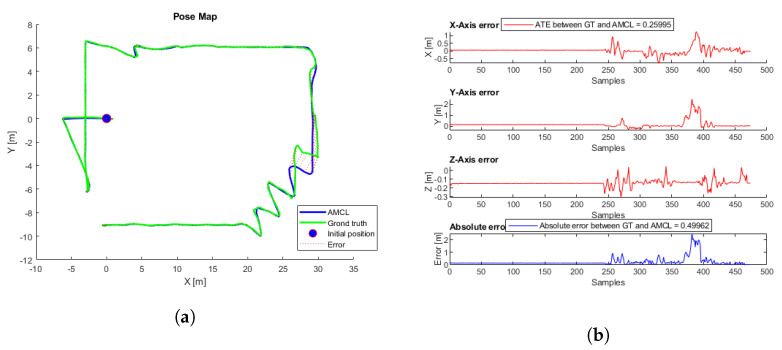
ICP-NL simulation test with the lowest ATE. (**a**) Trajectories. ICP-N (blue), ground truth (green), take-off position (blue circle), error (red). (**b**) ICP-N errors.

**Figure 30 sensors-22-05122-f030:**
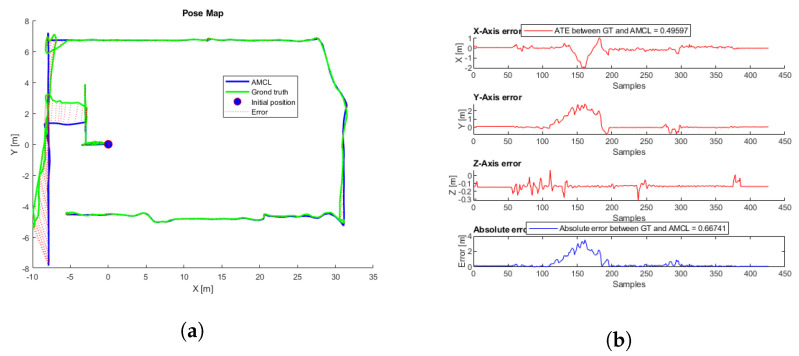
ICP-NL simulation test with the highest ATE. (**a**) Trajectories. ICP-NL (blue), ground truth (green), take-off position (blue circle), error (red). (**b**) ICP-N errors.

**Figure 31 sensors-22-05122-f031:**
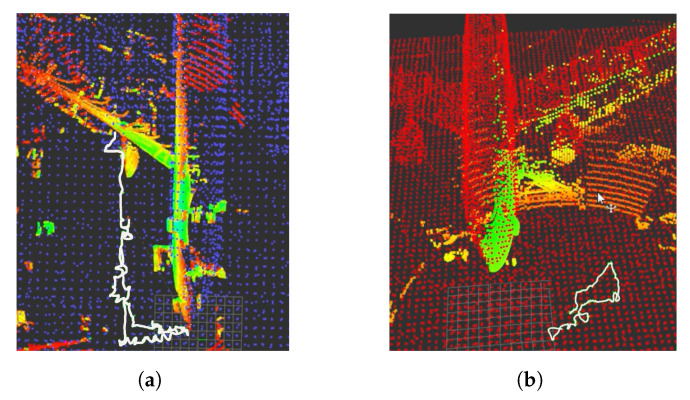
Performance results of the 3D localization algorithm with real data. Position estimate by the algorithm (white), 3D point cloud (green-yellow). (**a**) Path 1 test (Figure 16a). 3D graph SLAM map (blue). (**b**) Path 2 test (Figure 16b). 3D graph SLAM map (red).

**Table 1 sensors-22-05122-t001:** Calculation of ATE at different altitudes for 20 simulated flights.

Altitude (m)	3	4	5	6	7	8
**Mean ATE (m)**	0.3460	0.3287	0.3127	0.3376	0.4077	2.1437

**Table 2 sensors-22-05122-t002:** Calculation of minimum, maximum, and average ATE for 81 simulated flights.

Tests	ATE Min (m)	Ate Max (m)	Mean ATE (m)
81	0.18	5.63	0.34

**Table 3 sensors-22-05122-t003:** Initial positions for lowest and highest ATE trajectories.

ATE	GT Initial Pos (x,y) (m)	AMCL Initial Pos (x,y) (m)	Initial Covariance (xx,yy) (m)	Initial Error (x,y) (m)
Min ATE 0.18	32.53, −27.83	32.91, −27.5	0.5, 0.5	−0.38, −0.33
Max ATE 5.63	33.72, −29	36.01, −26.56	0.5, 0.5	−2.29, −2.44

**Table 4 sensors-22-05122-t004:** Relationship between the ATE, covariance in x and y directions, and matches of occupied grids cells and 2D laser scan points.

Error between AMCL and Ground Truth (m)	Mean Covariance(x,y)	Mean Occupancy Grip Map	Mean 2D Laser Scan	Mean Relation Matches/Laser-Scan
<=0.3	0.0557	41,820	245,917.5	0.17005

**Table 5 sensors-22-05122-t005:** Initial positions for lowest and highest ATE trajectories in truck inspections.

ATE	GT Initial Pos (x,y) (m)	AMCL Initial Pos (x,y) (m)	Initial Covariance (xx,yy) (m)	Initial Error (x,y) (m)
Min ATE 0.86	20.66, 7.16	20.77 , 5.56	0.5, 0.5	−0.11, −1.6
Max ATE 1.4	−4.82, −5	−4.77, −4.6	0.5, 0.5	−0.05, −0.4

**Table 6 sensors-22-05122-t006:** Relationship between the ATE, covariance in x and y, and matches of occupied grid cells and 2D laser scan points in the truck inspection.

Error between AMCL and Ground Truth (m)	Mean Covariance (x,y)	Mean Occupancy Grip Map	Mean 2D Laser Scan	Mean Relation Matches/Laser-Scan
<=0.3	0.0115	18,037	76,847	0.2347

**Table 7 sensors-22-05122-t007:** Errors obtained in NDT and ICP-NL simulation tests.

Algorithm	ATE
ICP-NL	0.68
NDT	289.93

**Table 8 sensors-22-05122-t008:** Aircraft simulation position estimation errors for ICP-NL and NDT algorithms for different altitudes.

Algorithm	Height (m)	ATE (m)	Convergence Time (s)
ICP-NL	3.5	0.7265	2.3
	4.5	0.7077	2.8
	5.5	0.7442	4
	6.5	0.6857	3.1
	7.5	0.6334	1.6
	8.5	0.552	4.1
NDT	3.5	149.6588	181.4
	4.5	511.6744	57.7
	5.5	363.2043	103.3
	6.5	133.5604	50.9
	7.5	0.6597	7
	8.5	1.6485	21.5

**Table 9 sensors-22-05122-t009:** Truck simulation position estimation errors for ICP-NL algorithms for different altitudes.

Algorithm	Height (m)	ATE (m)	Convergence Time (s)
ICP-NL	2.5	0.3801	2.3
	3.5	0.5552	2.8
	4.5	0.5442	4
	5.5	0.6857	3.1

## Data Availability

Not applicable.

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
