# Peer review of "Performance Analysis of Localization Algorithms for Inspections in 2D and 3D Unstructured Environments Using 3D Laser Sensors and UAVs"

_sensors, 2022, doi:10.3390/s22145122_

Round 1

Reviewer 1 Report

This paper presents “Performance analysis of localization algorithms for inspections in 2D and 3D unstructured environments using 3D laser sensors and UAVs”. The topic and application sound very interesting. However, some points should be addressed before considering this work as a possible publication.

1.       Some figures in section 2.0.2 are incorrect, e.g., in line 124 the correct Figure is the 2a. Similarly in line 133.

2.       Figure 3c should be changed to one with better resolution.

3.       Line 200 is missing parenthesis and a space after "(Figure 4”.

4.       The explanation in section 2.0.6 can be improved with a flowchart or an algorithm.

5.       The idea expressed in lines 236 and 237 is not clear.

6.       Sentence “The algorithm does not converge because the initial covariance is smaller than the error in the initial position” must appear in the descriptive text and not in the footer of Figure 8.

7.       Figure 12a is not mentioned in the text.

8.       It is recommended to numerate the videos.

9.       The conclusions are too long. It is recommended to place all the points discussed in a discussion section and finally, in the conclusions section, only place the relevant aspects of the work. The novelty and contributions of the work must be clarified in the conclusions section.

Author Response

Madrid, 13th June 2022

Reviewer 1 Sensors

Special Issue " Aerial Robotics: Navigation and Path Planning"

Subject: Response to Reviewer 1 Comments

Thanks for the time to analyze our paper. I must express my gratitude for your insightful comments and suggestions made. The comments have served to improve and complement the research and emphasize the quality of presentation and dissemination of the work.

The comments were responded at the end of this letter, describing the changes we have made and explaining the related problems' relevant aspects.

Sincerely,

Henry Paúl Espinosa Peralta

PhD. student in Automation and Robotics

Universidad Politécnica de Madrid

C/José Gutiérrez Abascal, 2.

28006 Madrid-Spain

https://www.car.upm-csic.es/

  1. Some figures in section 2.0.2 are incorrect, e.g., in line 124 the correct Figure is the 2a. Similarly in line 133.

Thanks to the comment, we updated the correct numbers of figures in lines 127, and 138 in the update document.

 We highlighted with green color the correction, and added a note.

  1. Figure 3c should be changed to one with better resolution.

Thanks for the observation. Each illustration of Figure 3, in line 163 were replaced respectively  by images of the aircraft model in the updated document

 We highlighted with green color the correction, and added a note.

  1. Line 200 is missing parenthesis and a space after "(Figure 4”.

The reviewer's suggestion has been considered. The correction is in line 205 (Figure 5) in the updated document.

    We highlighted with green color the correction, and added a note.

  1. The explanation in section 2.0.6 can be improved with a flowchart or an algorithm.

It is an excellent suggestion. We added pseudocode in line 242 in the updated document.

We highlighted with green color the correction and added a note.

  1. The idea expressed in lines 236 and 237 is not clear.

Thank you very much for the reviewer's suggestion. The idea was modified in line 244 in the updated document.

We highlighted with green color the correction and added a note.

  1. Sentence “The algorithm does not converge because the initial covariance is smaller than the error in the initial position” must appear in the descriptive text and not in the footer of Figure 8.

The reviewer's suggestion has been considered. We moved the text to lines 276 inside of Figure 10 in the updated document.

We highlighted with green color the correction and added a note.

  1. Figure 12a is not mentioned in the text.

Thanks for the comment. Figure 14.a was referenced in line 303 with respective descriptions in the updated document.

We highlighted with green color the correction and added a note.

  1. It is recommended to numerate the videos.

Thanks for the observation. We enumerated each video.

Line 169, 265 Video 1.  Line 311 Video 2. Line 314 Video 3. Line 218, 335 Video 4. Line 351 video 5. Line 352 video 6 in the updated document.

 We highlighted with green color the correction and added a note.

  1. The conclusions are too long. It is recommended to place all the points discussed in a discussion section and finally, in the conclusions section, only place the relevant aspects of the work. The novelty and contributions of the work must be clarified in the conclusions section.

Thank you very much for the reviewer's suggestion. We added a discussion section in line 359 and rewritten conclusions section in line 406.

We highlighted with green color the correction and added a note.

Reviewer 2 Report

The paper if very well written, easy to read, and extremely well presented. The only suggestion I have for the authors is to add two recent works, which uses very different approaches, for performing slam:

- Avola, D., Cinque, L., Fagioli, A., Foresti, G.L., Massaroni, C., Pannone, D. (2019). Feature-Based SLAM Algorithm for Small Scale UAV with Nadir View. In: Ricci, E., Rota Bulò, S., Snoek, C., Lanz, O., Messelodi, S., Sebe, N. (eds) Image Analysis and Processing – ICIAP 2019. ICIAP 2019. Lecture Notes in Computer Science(), vol 11752. Springer, Cham. https://doi.org/10.1007/978-3-030-30645-8_42

- Y. Luo, Y. Li, Z. Li and F. Shuang, "MS-SLAM: Motion State Decision of Keyframes for UAV-Based Vision Localization," in IEEE Access, vol. 9, pp. 67667-67679, 2021, doi: 10.1109/ACCESS.2021.3077591.

Author Response

Madrid, 13th June 2022

Reviewer 2 Sensors

Special Issue " Aerial Robotics: Navigation and Path Planning"

Subject: Response to Reviewer 2 Comments

Thanks for the time to analyze our paper. I must express my gratitude for your insightful comments and suggestions made. The comments have served to improve and complement the research and emphasize the quality of presentation and dissemination of the work.

The comments were responded at the end of this letter, describing the changes we have made and explaining the related problems' relevant aspects.

Sincerely,

Henry Paúl Espinosa Peralta

PhD. student in Automation and Robotics

Universidad Politécnica de Madrid

C/José Gutiérrez Abascal, 2.

28006 Madrid-Spain

https://www.car.upm-csic.es/

  1. The paper if very well written, easy to read, and extremely well presented. The only suggestion I have for the authors is to add two recent works, which uses very different approaches, for performing slam:

- Avola, D., Cinque, L., Fagioli, A., Foresti, G.L., Massaroni, C., Pannone, D. (2019). Feature-Based SLAM Algorithm for Small Scale UAV with Nadir View. In: Ricci, E., Rota Bulò, S., Snoek, C., Lanz, O., Messelodi, S., Sebe, N. (eds) Image Analysis and Processing – ICIAP 2019. ICIAP 2019. Lecture Notes in Computer Science(), vol 11752. Springer, Cham. https://doi.org/10.1007/978-3-030-30645-8_42

- Y. Luo, Y. Li, Z. Li and F. Shuang, "MS-SLAM: Motion State Decision of Keyframes for UAV-Based Vision Localization," in IEEE Access, vol. 9, pp. 67667-67679, 2021, doi: 10.1109/ACCESS.2021.3077591.

The reviewer's suggestion has been considered. Cites are included in the paragraph about slam algorithms in lines 61 and 67 in the updated document.

Reviewer 3 Report

In this paper, the authors present two different localization methods for aircraft inspection. There are several comments:

  1. The written and organization of "Introduction" is bad. This reviewer suggests to rewrite the Introduction. This is also the main concern about this manuscript. From the introduction, the reader cannot understand what work you want to do, and why you do this work, and what is the key problem that you want to solve. The reviewer suggests to rewrite it with a clear clue.
  2. The authors should add a section "related work" to review the related papers.
  3. Fig.3, The reviewer suggests to use the model of aircraft to illustrate the issue of grid map construction.
  4. Section 2.0.5, please add a figure to show the result of 2D localization
  5. Section 3, please compare your methods with several existing localization approaches.
  6. I would be better if the authors can provide the other examples. In Abstract, the authors declare " We focus on aircraft inspection as a test example, but our results and conclusions can be directly extended to other applications" . This reviewer think that it should be proved using an experiment. 

Author Response

Madrid, 13th June 2022

Reviewer 3 Sensors

Special Issue " Aerial Robotics: Navigation and Path Planning"

Subject: Response to Reviewer 3 Comments

Thanks for the time to analyze our paper. I must express my gratitude for your insightful comments and suggestions made. The comments have served to improve and complement the research and emphasize the quality of presentation and dissemination of the work.

The comments were responded at the end of this letter, describing the changes we have made and explaining the related problems' relevant aspects.

Sincerely,

Henry Paúl Espinosa Peralta

PhD. student in Automation and Robotics

Universidad Politécnica de Madrid

C/José Gutiérrez Abascal, 2.

28006 Madrid-Spain

https://www.car.upm-csic.es/

  1. The written and organization of "Introduction" is bad. This reviewer suggests to rewrite the Introduction. This is also the main concern about this manuscript. From the introduction, the reader cannot understand what work you want to do, and why you do this work, and what is the key problem that you want to solve. The reviewer suggests to rewrite it with a clear clue.

It is an excellent suggestion. We rewrite the introduction to be more clear. The text is in lines 27 to 55.

We highlighted with green color the correction and added a note.

  1. The authors should add a section "related work" to review the related papers.

Thank you very much for the reviewer's suggestion. We added a section “Related work” in lines 56 to 107.

We highlighted with red color the correction and added a note.

  1. 3, The reviewer suggests to use the model of aircraft to illustrate the issue of grid map construction.

Thanks for the observation. Each illustration of Figure 3 in line 163 was replaced respectively by images of the aircraft model in the updated document.

We highlighted with yellow color the correction and added a note.

  1. Section 2.0.5, please add a figure to show the result of 2D localization

Thank you very much for the reviewer's suggestion. An example of 2D localization was added in Figure 4 line 194 and referenced in Video 1 in line 169. An example of 3D localization was added in Figure 6 line 220 and referenced in Video 4 in line 218 in the updated document.

We highlighted with blue color the correction and added a note.

However, Figure 8 line 265 shows some steps of the convergence algorithm in 2D localization, and Figure 20 line 335 shows 3D localization by Non-linear ICP and NDT in the updated document.

  1. Section 3, please compare your methods with several existing localization approaches.

Thank you very much for the reviewer's suggestion. But our paper focuses on the analysis and performance of specific 2D and 3D algorithms using a map previously created from a CAD model. Therefore, comparing performance with other algorithms is not applicable in this analysis.

Concerning 2D algorithms, we use AMCL as an alternative to other localization methods, which require an initial position with a low value of uncertainty error to converge. The results of tests performed with difertent position values are shown in Figure 9 and Figure 10 in line 279, additionally in Table 3 in line 293 we show data of correct and incorrect trajectory estimation when performing a flight. One  conclusion is to use the estimated position of the AMCL algorithm when it has converged as the starting position for other algorithms, such as EKF.

In addition, we have made a comparison between two 3D localization algorithms ICP-NL and NDT to observe which one presents a better position estimation when starting flights with different altitudes and initial positions as shown in Table 6 line 343, and Figure 21 and 22 in line 340.

  1. I would be better if the authors can provide the other examples. In Abstract, the authors declare " We focus on aircraft inspection as a test example, but our results and conclusions can be directly extended to other applications". This reviewer think that it should be proved using an experiment. 

It is an excellent suggestion. We performed the analysis on aircraft inspections as an example. But the methodology, map creation, software, and hardware tools can be used in other scenarios where there are objects of considerable size to be inspected, such as trucks, wind towers, buildings, etc.

Our results and conclusions will be similar when using another environment. Because the algorithms tested in 2D and 3D are based on the finding of coincidences between the laser scanner and the created map. However, each environment may change the number of coincidences between the laser and the map and consequently the convergence time, this is a function of the relevant features found on the map.

Reviewer 4 Report

This study analyzes two different approaches on autonomous navigation and localization and relative positioning of Unmanned Aerial Vehicle’s (UAV) to for inspecting relevant objects in unstructured environments. The first technique, based on Adaptive Monte Carlo Localization (AMCL), focuses on 2D localization. The second technique focuses on 3D localization and is based on Iterative Closest Point algorithm (ICP), Normal Transformation Distribution (NDT) and Unscented Kalman Filter (UKF).

The introduction section of the manuscript is well written and documented, providing the background of the study. This section offers a good analysis about the localization methods of autonomous robotic systems during an autonomous inspection, in order to estimate the robot position considering the uncertainty in the sensors data and the vehicle dynamics.

The Methods section is presenting the case of aircraft inspection because it takes place in an unstructured environment and aircraft position may be different at each parking lot. For this work, an open 3D CAD airplane model was used. The selected methods and procedures would be similar in other inspection tasks and other UAV models. The section describes the types of UAVs, sensors and software used for both simulation and real flights. A map of the environment and an 3D Occupancy grid Map (Octomap) and an 2D Occupancy grid Map were built from the CAD model to test the performance of the algorithms.

The Absolute Trajectory Error (ATE) was used for robot trajectory estimation. Localization in 2D is performed by means of the AMCL algorithm and trajectories with the smallest error were selected after simulation flights. 3D localization is performed by adapting the NDT and IMU data with UKF algorithm, taking as inputs the 3D point cloud map of the environment, the previous state including position and orientation, LIDAR sensor data and Inertial Measurement Units (IMU) data together with linear and angular velocities. The scan matching algorithm performs the pose estimation by aligning the map point cloud and the sensor point cloud.

The Results section explains that a 3D CAD model in Digital Asset Exchange (DAE) format for Gazebo was imported in order to build the 2D occupancy grid map. An octomap of the environment was created, then it was projected on a horizontal flat plane to make a 2D occupancy map and used to obtain the image’s edges. The AMCL parameters were calibrated in simulation test flights to verify the localization quality. When the UAV is flying longer, the algorithm converges and the 2D laser scan matches the map.

Real flights were executed before starting the inspection to check the algorithm’s convergence. In order to obtain better results, walls were removed by limiting the 2D laser scan data in range and angle of view.

The 3D map was built based on the graph SLAM algorithm odometry obtained from NDT-scan matching with corrections based on GPS data. The ATE has been calculated for the flights performed and the NDT and ICP-NL algorithms have been compared to observe their performance and differences. The NDT algorithm tries to estimate a trajectory but it fails to converge. The algorithm ICP-NL converges during take-off because the sensor observes more points, and during navigation the error remains low.

In order to study the influence on the results several flights were performed by modifying the UAV flying height. ICP-NL maintains its performance regardless of the altitude and is faster to converge. NDT works well in higher altitudes because the aircraft is observed better. The trajectories generated by the ICP-NL algorithm for the simulated flight and the ground-truth are practically overlapped along the whole path.

Tests performed with real data  show that the trajectory estimated by ICP-NL has many deviations, and the 3D map data does not match the 3D laser data because of the noise generated by the walls and adjacent objects.

The Conclusion section shows that for 2D Localization using a static 2D map built from a standard CAD model has significant advantages over developing SLAM algorithms to map the environment, can be cost-effective, as it can save time and resources with good results. Unexpected objects not present in the map can be segmented in a simple manner by modifying the laser data range. When using AMCL the algorithm can estimate the location accurately after some time and can be used to determine a reliable initial position with little error by performing a specific trajectory and waiting for the algorithm to converge.

For 3D Localization the graph SLAM map the Graph-Slam changes applied for this work were adequate. NDT can achieve high accuracy in the estimation of the X and Y coordinates in some of the flights but the proposed ICP-NL scan-matching method and UKF performed better in the performed tests.

The paper showed that a map generated in simulation can be used for localization in a real environment with good results.

Author Response

Madrid, 13th June 2022

Reviewer 4 Sensors

Special Issue " Aerial Robotics: Navigation and Path Planning"

Subject: Response to Reviewer 4 Comments

Thanks for the time to analyze our paper. I must express my gratitude for your insightful comments and suggestions made. The comments have served to improve and complement the research and emphasize the quality of presentation and dissemination of the work.

Sincerely,

Henry Paúl Espinosa Peralta

PhD. student in Automation and Robotics

Universidad Politécnica de Madrid

C/José Gutiérrez Abascal, 2.

28006 Madrid-Spain

https://www.car.upm-csic.es/

Round 2

Reviewer 3 Report

1) The structure of Section 3 is unreasonable.  For example, subsection 3.0.4 should be in Section 4.  The reviewer also think that the discriptions of Software and Hardware should not be presented in Section Methodology. Because the Section Methodology should only  describe your methods. The Software and Hardware setting should be presented in Experiments. 

2) The authors should illustrate how and where to collect the real data.

3) The reviewer still think the experiments conducted on aircraft inspection are not enough to support the claim "our results and conclusions can be directly extended to other applications " made in Abstract.  The authors can use another simulated dataset, such as trucks and wind towers, to support this claim. 

Author Response

Madrid, 30th June 2022

Reviewer 3 Sensors

Special Issue " Aerial Robotics: Navigation and Path Planning"

Thanks for the time to analyze our paper. I must express my gratitude for your insightful comments and suggestions made. The comments have served to improve and complement the research and emphasize the quality of presentation and dissemination of the work.

Sincerely.

Henry Paúl Espinosa Peralta

PhD. student in Automation and Robotics

Universidad Politécnica de Madrid

C/José Gutiérrez Abascal, 2.

28006 Madrid-Spain

https://www.car.upm-csic.es/

Response to Reviewer 3.

  • The structure of Section 3 is unreasonable.  For example, subsection 3.0.4 should be in Section 4.  The reviewer also think that the discriptions of Software and Hardware should not be presented in Section Methodology. Because the Section Methodology should only describe your methods. The Software and Hardware setting should be presented in Experiments. 

Thank you very much for the reviewer's suggestion.

We changed software, hardware, and ATE error at the beginning of “Section Results” in lines 221 to 250, Also we add a short descriptive text to guide the reader in line 218.

Text highlighted in yellow.

  • The authors should illustrate how and where to collect the real data.

Thanks for the observation.

We added a description of how we collected the data into “Section 4.1.2. Real tests” in lines 317 to 320. Also, figure 16 shows trajectories realized in the real environment.

Text highlighted in green.

  • The reviewer still think the experiments conducted on aircraft inspection are not enough to support the claim "our results and conclusions can be directly extended to other applications " made in Abstract.  The authors can use another simulated dataset, such as trucks and wind towers, to support this claim. 

It is an excellent suggestion.

We performed other simulations using a 3D truck, we used the same methodology and configuration parameter in 2D and 3D simulations.

 In lines 301 to 316, we present results of 2D localization. We executed several flights to find the lowest and highest ATE (Figure 14, 15). Also, video 7 shows a test localization. Table 5 shows data about take-off position and initials parameters for lowest and highest ATE. Table 6 shows data Covariance XY, and the relation between occupied cell and 2D laser scan to determine when the algorithm converges. We can observe these values are lower than aircraft inspection because the map is smaller. These values depend on the size map and size object to inspect.

 In lines 377 to 386, we present results of 3D truck localization. We executed several flights to find the lowest and highest ATE (Figure 29, 30). Also, video 9 shows a test localization. These data and video 9 allow the observer a good pose localization using ICP-NL.

Text highlighted in cyan.
